# Distress Levels of Parents of Children with Neurodevelopmental Disorders during the COVID-19 Pandemic: A Comparison between Italy and Australia

**DOI:** 10.3390/ijerph182111066

**Published:** 2021-10-21

**Authors:** Dayle Burnett, Anne Masi, Antonio Mendoza Diaz, Renata Rizzo, Ping-I Lin, Valsamma Eapen

**Affiliations:** 1Department of Women’s and Children’s Health, Uppsala University, 752 36 Uppsala, Sweden; dayle.burnett-quarry.11@alumni.ucl.ac.uk; 2School of Psychiatry, University of New South Wales, Kensington 2052, Australia; a.masi@unsw.edu.au (A.M.); a.mendozadiaz@unsw.edu.au (A.M.D.); 3Academic Unit of Child Psychiatry, South Western Sydney Local Health District & Ingham Institute, Sydney 2170, Australia; 4Department of Clinical and Experimental Medicine, University of Catania, 95124 Catania, Italy; rerizzo@unict.it; 5Mental Health Research Unit, South Western Sydney Local Health District, Liverpool 2170, Australia

**Keywords:** COVID-19, pandemic, parental distress, neurodevelopmental disorder

## Abstract

Parents of children with a neurodevelopmental disorder (NDD) report higher levels of distress compared to those of typically developing children. Distress levels may be heightened by the restrictions associated with the COVID-19 pandemic. However, it is unclear whether distress levels of parents varied by the diagnosis of neurodevelopmental disorder in children during the COVID-19 pandemic. This study aims to investigate whether parental distress was influenced by the type of NDD. Participants were from Australia (*N* = 196) and Italy (*N* = 200); the parents of children aged 3–18 were invited to complete an online self-reported survey which included the 6-item Kessler Psychological Distress Scale (K6) to determine parental distress. The results show that intellectual or learning disorder (ILD) is a major contributor to parental distress compared to other NDDs in both Australia and Italy. Moreover, the worsening of symptomatic changes in children with NDDs was significantly associated with parental distress. The differences between the two countries in terms of the pandemic impact, however, were not statistically significant. The results suggest that intervention strategies need to be tailored for individual clinical information and factor in the society’s stringency level of anti-contagion policies to improve parental wellbeing.

## 1. Introduction

The COVID-19 pandemic has led to changes in many different aspects of life around the world during the year 2020. Its adverse consequences have not completely subsided in 2021 due to subsequent waves of infections and variants that lead to repeated anti-contagion measures including social distancing, home quarantine, and the closure of schools. Although these measures are effective in curbing the spread of the virus, the disruption of services, social isolation, and financial insecurities have caused a huge psychological impact worldwide [1]. In particular, a high prevalence of psychological symptoms, such as anxiety and depression, has been reported in parents of young children and adolescents [2,3].

During home quarantine, family daily routines unexpectedly changed, which changed the role of parents. After the closure of schools, most parents had to combine working from home and childcare, while simultaneously homeschooling their children. Moreover, the economic implications of the COVID-19 have also been shown to negatively impact families [4,5]. The disruption of daily routines may be even more difficult for children with neurodevelopmental disorders (NDDs), as established routines are a crucial strategy to promote stability [6]. In addition, the closure of educational or other therapeutic placements meant that children with NDDs lost the professional support they need [7,8].

In general, parents of children with NDDs experience higher levels of parental distress compared to those of typically developing children [9,10]. In addition, they are more likely to experience mental health problems such as anxiety, depression and reduced sleep quality [11,12,13]. Findings from existing research suggest factors such as social support, children’s behavioural problems, economic and social status and parenting stress are key factors that contribute to psychological distress in parents of children with NDDs. Studies have also suggested that parents of children with autism spectrum disorder (ASD) report higher levels of parental distress compared to other NDDs [14,15].

Based on the above research, it can be expected that the COVID-19 pandemic and related restrictions to routine are likely to have a negative impact on parents of children with NDDs. Indeed, emerging evidence supports this notion; for example, data from our Australian survey of parents of children with NDDs found that COVID-19 had an adverse impact on the wellbeing of three-quarters of parents while over 40% reported worsening of their pre-existing mental health issues [16]. This appears to be compounded by the fear of infection and the impact of lockdowns and consequent social isolation as well as adverse economic consequences. Similarly, a study in Italy using the same survey found that 58.5% of respondents found that their child’s overall health and wellbeing had been impacted by the pandemic, while 47.7% stated their own wellbeing had been affected [17].

Despite the relatively few studies investigating the impact of the COVID-19 pandemic on the wellbeing of parents of children with NDDs, it is clear that these parents need support. Clarifying the clinical and sociodemographic contributors to parental psychological wellbeing may facilitate the implementation of mental health for families with children with NDDs [18]. To further explore the relationship between parental distress and children with NDDs, this study aimed to address. The study aims to address two primary research questions: (1) Which diagnosis of NDD was more likely to correlate with higher levels of parental distress during the COVID-19 pandemic? (2) Which diagnosis-specific symptomatic change was more likely to correlate with parental distress levels during the COVID-19 pandemic? We believe that the answers to these questions might pave the way for targeted prevention or intervention strategies for families with special needs. Further, we aimed to investigate how the magnitude of the disease burden of COVID-19 in society as a whole might impact parental distress differently. From January to June 2020, Australia was in the top 10% of countries with the lowest infection and mortality rates related to COVID-19, while Italy was one of the top 10% countries with the highest infection and mortality rates related to COVID-19 [19]. The disease burden has led to variable levels of stringency index values (an indicator for how strictly non-pharmaceutical intervention measures are implemented) across different countries [20].

## 2. Methods

### 2.1. Research Design

A cross-sectional design was used to investigate the impact of the COVID-19 pandemic on Italian and Australian families. Parents and caregivers of a child aged 3–18 years with an NDD were asked to participate in the surveys. The survey was promoted via disability service providers and support groups by emailing parents on their mailing lists with information about the study and the link to the survey, or by posting an advertisement with the link on social media. Interested parents could read the Participant Information Statement online and complete the survey via Research Electronic Data Capture (REDCap), a secure web-based survey tool [21,22]. Consent to participate was considered implied if parents elected to complete the survey. The survey was open for a period of approximately 6 weeks during May and June 2020.

### 2.2. Measures

The survey consisted of questions about the participants’ socio-demographic characteristics and the relevant diagnoses the child had received, along with a rating of symptom change since the start of the pandemic. A 5-point scale with 1 = “symptoms much improved”, 2 = “symptoms somewhat improved”, 3 = “symptoms the same”, 4 = “symptoms somewhat worse” and 5 = “symptoms much worse” was used. We combined responses from 4 = “symptoms somewhat worse” and 5 = “symptoms much worse” into one group indicating worsening of the child’s condition. In addition, there were questions relating to: general wellbeing (perceived impact of the pandemic across the family), family health (employment, finances, food and housing), home-based learning (challenges in homeschooling) and child behaviors (including emotional responses and use of technology and devices). The above questions were rated on a 5-point Likert scale ranging from “strongly agree” to “strongly disagree”. Similarly, the two responses at the impacted end of the scale—”strongly agree” and “somewhat agree” were combined together to represent a negative impact felt by the respondents. Parental distress was assessed using the Kessler Psychological Distress Scale (K6), a robust measure of psychological distress for adult populations [23]. The K6 consists of 6 questions that ask about the frequency that participants felt sad, nervous, restless, hopeless, that everything was an effort, and worthless during the past 30 days.

### 2.3. Statistical Analysis

We first compiled the results of the survey for Italy and Australia separately to allow a direct comparison between the two countries. We compared the distributions of demographic and clinical variables between the two countries by carrying out chi-squared tests. Regression models were constructed to evaluate our hypotheses to determine which NDD diagnosis in the child can predict parental distress in Australia and Italy, as well as in the total population by merging the Australia and Italy datasets. Furthermore, we used semi-partial correlation coefficients to infer the proportion of variance in parental distress levels explained by each NDD to compare the relative contribution of each diagnosis. The two-sided alpha value = 0.05 was used to determine the relationship between the independent variable and the dependent variable.

To prove that the two countries had different stringency index levels in 2020, we extracted the data from “Our World in Data—COVID-19 section” (https://ourworldindata.org/coronavirus, accessed on the 1 August 2020) and compared the stringency index values between Italy and Australia, and confirmed that the stringency index values statistically significantly varied between these two countries using a mixed model analysis (*p* < 0.0001). Therefore, we also compared the impacts on parental distress between Australia and Italy, the two countries that represent two groups of countries affected by the COVID-19 pandemic with different levels of stringency index values.

Next, we carried out interaction analyses to compare the impact of the child’s diagnosis on parental distress between Australia and Italy. In addition, interaction analyses were also used to investigate whether the impact of diagnosis-specific symptomatic changes on parental distress varied by country. In the linear regression model, the outcome variable (i.e., Kessler-6 scores) was regressed against the primary predictor (i.e., either diagnosis or diagnosis-specific symptomatic change), country, and the product of the primary predictor and the country, adjusting for the parental age and educational level. All analyses were conducted in R version 4.00(R Core Team (2020). R: A language and environment for statistical computing. R Foundation for Statistical Computing, Vienna, Austria) [24].

## 3. Results

### 3.1. Overview of the Samples

A total of 200 respondents participated in the survey in Italy, while 296 respondents participated in Australia. As shown in Table 1, the caregivers participating in the study were mostly between the ages of 40 and 49 in both Italy (53.0%) and Australia (54.8%). The mean age of the participants’ children was 10.7 and 10.1, in Italy and Australia, respectively, with the majority of the children male in both Italy (81.5%) and Australia (65.9%). In Australia, a higher percentage of caregivers had higher education compared to Italy (68.8% vs. 19.5%).

Table 2 provides an overview of the NDD groups. In Italy, the most common NDDs were Tourette’s syndrome (TS) (54.5%), Autism Spectrum Disorder (ASD) (51.0%) and Intellectual or Learning Disorder (ILD) (37.5%). In Australia, the most common NDDs were ASD (61.3%), ADHD (40.7%) and Anxiety Disorder (33.7%). Parents were also asked whether there were any symptomatic changes in their children during the pandemic. Although significant differences in the symptomatic changes were observed between Italy and Australia across all NDD groups, at least half of the respondents in both Italy and Australia experienced worsening of the symptoms in children with ADHD (50.0% vs. 58.7%), Anxiety Disorder (50.0% vs. 68.0%), OCD (51.4% vs. 51.6%) and TS (52.3% vs. 64.3%). In Australia, children with ASD were perceived to have increased symptoms (58.0%) compared to children in Italy (37.3%). Less than half of the respondents in both Italy and Australia experienced worsening of the symptoms in children with ILD (42.7% vs. 37.5%).

The K6 scores of the parents are presented in Table 3. On average, parents in Italy showed significantly higher distress levels compared to parents in Australia. For example, 36.0% of the respondents reported being nervous most or all of the time in Italy, whereas only 20.3% felt the same way in Australia. Similarly, 26.0% of parents in Italy reported that most or all of the time they felt so depressed that nothing could cheer them, while only 9.1% felt the same way in Australia. However, when asked “how often did you feel that everything was an effort?”, a greater percentage felt that way most or all the time in Australia, 25.7%, compared to only 20.0% of parents in Italy.

The following results can be found in the Appendix A. More than half of the participants in both Italy and Australia somewhat or strongly agreed that their child’s overall health and wellbeing had been impacted by the pandemic with 58.5% and 69.4%, respectively. However, just under half (45.5%) believed the pandemic had worsened preexisting health conditions for their child in Italy, with half of the respondents feeling the same way in Australia.

With regards to support networks, caregivers in Australia were impacted more than those in Italy. For example, 77.4% of respondents felt their support networks had decreased compared to 53.0% in Italy. Similarly, 71.8% felt that the pandemic had disrupted caregivers’ support and services in Australia, whereas only 44.89% felt the same way in Italy. Almost three-quarters of the respondents in Italy and Australia, 74.3% and 72.2%, respectively, felt that the pandemic has significantly disrupted the allied health services their child accessed. Most of the respondents in Australia (97.1%) stated that their children took more medication than normal. In Italy, a smaller proportion (29.2%) of caregivers felt that their child’s ability to access specialists had been significantly impacted by the pandemic compared to a much larger proportion of caregivers in Australia (70.5%).

With regards to contributors to overall family health, caregivers in Australia were more impacted than those in Italy with 92.9% stating that COVID had significantly disrupted their child’s routines compared to 69.0% in Italy. Similarly, 75.5% indicated that during the past two weeks, COVID restrictions had been stressful for their child compared to 47.0% in Italy. Moreover, when asked about balancing work with childcare and family responsibilities, the majority (80.9%) of caregivers in Australia reported it had been difficult, whereas less than half (40.0%), felt the same way in Italy. Nearly all the respondents in Italy were optimistic the COVID crisis would end soon (94.5%) with just under three-quarters (73.2%) feeling the same in Australia.

The pandemic had a significant impact on the children’s education in both Italy and Australia, with respondents stating that COVID-19 had prevented their child from attending school with 90.3% and 91.1%, respectively. Although most of the respondents had adequate access to online resources, fewer than half in both Italy (47.5%) and Australia (44.4%) felt their child had adequate capacity to engage in home-based learning. Interestingly, the majority of caregivers in Italy (79.9%) felt that they had adequate capacity to support their child’s educational needs, whereas less than half (46.4%) felt the same way in Australia.

A proportion of caregivers in Italy reported a reduction in sleep quality in their children (18.7%) compared to almost half (46.9%) of caregivers in Australia. In both Italy and Australia, more than half of the respondents reported that their child had become more easily annoyed since the start of the pandemic (Italy: 60.6% vs. Australia 65.2%). Moreover, the majority of the parents reported that their children spent more screen time and played significantly more video games since the outbreak (Italy: 58.7% vs. Australia 65.4%).

### 3.2. Impact of Diagnoses of Children on Parental Distress

The results from the linear regression analysis in Australia (Table 4) show the diagnosis of ILD in children is significantly positively correlated with parental distress after adjusting for the other NDDs, the child’s age, the caregivers’ age and education level. Interestingly, parents with children diagnosed with TS were significantly inversely correlated with parental distress levels during the pandemic after adjusting for the same confounders. Although there were no significant associations between NDDs and parental distress in Italy, children with ASD showed a positive correlation with parental distress. When merging the two datasets together, it became apparent that ILD could be a major contributor to parental distress during the pandemic (*B* = −1.2654, *p* = 0.038). However, children diagnosed with TS were inversely correlated with parental distress (*B* = 2.7612, *p* < 0.005). The proportion in the K6 score explained by each diagnosis of the child in each country is shown in Figure 1. The results indicate that, among all diagnoses, the largest contributor to parental stress in Australia was TS, while the largest contributor to parental stress in Italy was ASD.

Figure 2 provides an overview of the interaction plots which illustrates how Italian and Australian parents experienced different levels of stress during the pandemic by the following child diagnoses: ASD, TS, ILD and OCD. The results confirm the earlier analyses that children diagnosed with TS showed lower parental distress levels compared to those without the diagnosis. There was no significant difference between Australia and Italy regarding the impact of NDDs on parental distress (Table 5).

In Italy, higher parental distress levels were significantly associated with the worsening of the symptomatic changes in children with ILD, OCD and TS—which is also shown in Table 6. In Australia, only worsening of the symptoms in ASD or ILD was associated with increased parental distress levels, while worsening of symptoms in OCD and TS did not statistically significantly correlate with parental distress levels. These results imply that the impact of symptomatic changes in OCD or TS on parental distress might vary by country during the pandemic. However, none of the interaction effects between the diagnosis-specific symptomatic changes and the country on parental distress reached statistical significance levels (Table 7). Figure 3 shows the results of the effect of diagnosis-specific symptomatic change and its effect on parental distress.

## 4. Discussion

In this study, we investigated the predictors of parental distress pertaining to NDDs and diagnosis-specific symptomatic changes. Moreover, we aimed to compare the impact of COVID-19 factors between Italy and Australia.

The results from this study provide important insights into the predictors of parental distress during the pandemic and suggest that ILD in children is a major contributor in both Australia and Italy. This is in accordance with two recent studies carried out in the UK which reported increased levels of mental health in caregivers of children with intellectual disabilities [25,26]. Children with ILD are more likely to require more face-to-face special educational support than other children and hence interruptions of such support during the pandemic may pose a greater challenge for parents compared with others. Our study did not show any significant results with regards to children diagnosed with ASD increasing parental distress during the pandemic. This is in accordance with a recent paper that evaluated parenting stress before and during the pandemic in children with NDDs and their families [27]. However, our finding is contrary to previous studies that reported increased distress levels of parents of children with ASD during the pandemic [28,29]. This result may be explained by the fact that our study only included parents of children with NDDs, whereas other studies compared parental distress levels of children with NDDs to those of typically developing children. Furthermore, the life routines of children with ASD, who have fewer needs for social activities than some non-ASD children, may be less likely to be affected by the pandemic compared with children without ASD. Therefore, social distancing could cause less stress among children with ASD than other non-ASD children although these children with ASD might perceive more stress when school-based services could become less accessible due to school closures. Taken together, the child’s diagnosis of ASD may not necessarily cause higher levels of distress compared with other non-ASD children with or without other NDDs.

One unanticipated finding from this study was the diagnosis of TS in children appeared to be associated with lower levels of parental distress when compared to other NDDs. A possible explanation is that parents or caregivers of children with TS might receive decreased unpleasant attention in public because of lockdowns during the pandemic era. Furthermore, children with TS might have less need for face-to-face special educational support than children with some other NDDs, such as ASD and ILD. Therefore, compared with families of children with various types of NDDs, homeschooling and other anti-contagion measures seemed to yield a limited impact on parents of children with TS. Indeed, previous research has speculated some positive aspects of home quarantine [30].

With respect to the symptomatic changes of children with NDDs, we found that the worsening of symptoms in children with ASD, ILD and TS were significantly correlated with increased levels of parental distress, which is consistent with recent findings [31,32] This could be explained by the impact of home quarantine disrupting children’s routines, given that most parents in both Italy and Australia stated that COVID significantly disrupted their child’s routines. It should be noted, however, that the levels varied between Australia and Italy. For example, in Australia, we observed high levels of parental distress in parents of children with OCD and TS irrespective of symptomatic changes. Although, it is unclear whether the difference in the impact of symptomatic changes associated with these two NDDs between Italy and Australia could be attributable to the level of stringency concerning anti-contagion measures at the country level.

Indeed, our study found many differences between Italy and Australia with regards to the impact of the pandemic. On average, parents in Italy showed higher distress levels compared to Australia based on the K6 score, while parents in Australia self-reported a higher impact on aspects such as types of support and the worsening of symptoms in children. The differences in Italy and Australia are most likely due to both countries being in different stages of the pandemic at the time of the survey. Other different parental features that impact parental distress levels may account for differences in pandemic’s impact between Italy and Australia (e.g., educational attainment in Italy and parent–child relationship in Australia) may demand further investigations using larger samples.

A number of limitations need due consideration while interpreting the findings. Firstly, this is a cross-sectional survey and hence the inter-relationships and directionality of the impact over time are unclear. This survey was conducted during the early phase of the pandemic in June/July 2020 and hence the long-lasting economic and social impact of the pandemic and the consequent effects on parental mental health and the parent–child interactions may not have been fully unfolded at that time. Secondly, parents were asked to participate in a survey which may have led to self-selection bias that might stem from any circumstances that distinguish the parents who participated in the survey from the other non-participating parents. Further, the survey was performed as a parent report and hence the self-report bias cannot be excluded. Thirdly, the modest sample size in both Italy and Australia was only intended to relate to a portion of the NDD community. Although cross-country comparisons could shed some light on the association between environmental factors and health issues, caution needs to be exercised in the interpretation of these findings for several reasons. First, multiple unmeasured confounders may be involved in differences between countries. Second, ecological fallacy may arise when we use aggregate data to infer individual data [33]. In the current study, we did not intend to use aggregate data to infer the relationship between parental distress and children’s features. However, the difference in parental distress levels in relation to children’s clinical features between the two countries may not be attributable to the difference in aggregate data, such as the magnitude of the disease burden associated with COVID-19 between these two countries. Finally, self-reported categorical data lack cross-population comparability [34]. Taken together, these reasons could limit the interpretations of our findings in the context of cross-country comparison. Future studies with larger representative samples are needed for ensuring the generalisability of the findings. Additionally, longitudinal data on changes in distress levels at multiple time points over the course of the pandemic may cast insights into the long-term impact of the pandemic on the wellbeing of parents of children with neurodevelopmental disorders.

## 5. Conclusions

The current study is the first to examine cross-country comparisons of parents of children with NDDs by comparing Italy (longer lockdowns and school closures in 2020) versus Australia (shorter lockdowns and school closures in 2020). The results indicate that the pandemic might yield different impacts on the psychological wellbeing of parents (or caregivers) of children with different types of NDDs, and such differences might also vary between Italy and Australia. Although it appears that the difference in how stringently anti-contagion measures were implemented might yield, at best, a limited impact on how parental wellbeing could be affected by NDDs in children, more research is warranted to investigate the relationship between the level of stringency of lockdowns and school closures parental wellbeing. Family and social support measures aligned with the unique characteristics of each vulnerable group need to be revised to cater to the unique needs of families of children with different NDD diagnoses.

## Figures and Tables

**Figure 1 ijerph-18-11066-f001:**
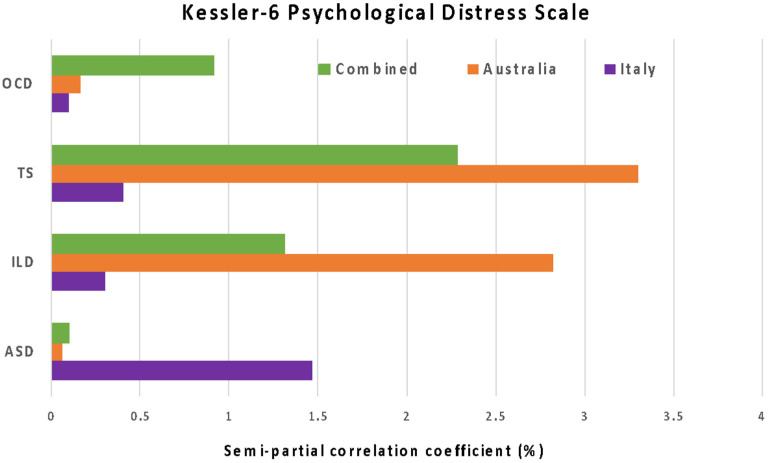
Proportions in the variation in parental stress explained by the child’s NDD diagnosis.

**Figure 2 ijerph-18-11066-f002:**
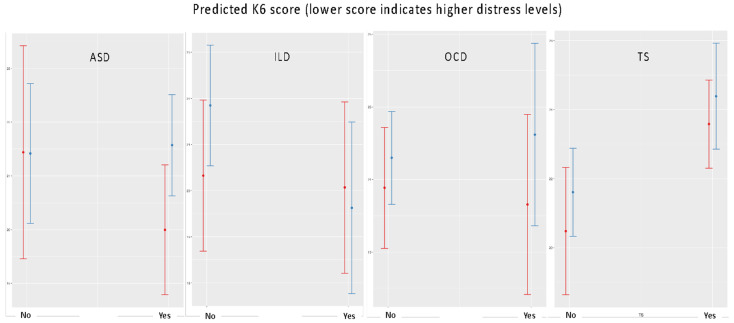
Parental distress levels stratified by the child’s diagnosis and country. Vertical bars (red and blue bars indicate Italian and Australian samples, respectively) refer to the 95% CIs.

**Figure 3 ijerph-18-11066-f003:**
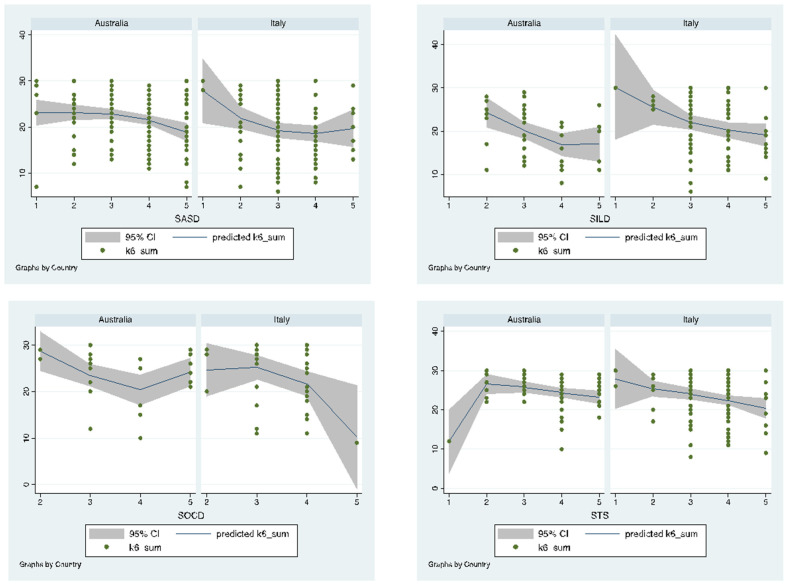
Comparing the effects of symptomatic change of NDDs on parental distress between Italy and Australia. SASD indicates the symptomatic change for ASD; SILD indicates the symptomatic change for ILD; SOCD indicates the symptomatic change for OCD; STS indicates the symptomatic change for TS. Lower K6_sum scores indicate higher distress levels. Higher scores in the symptomatic change indicate worsening of symptoms during the pandemic.

**Table 1 ijerph-18-11066-t001:** Demographic features of the two samples.

	Italy (*N* = 200)		Australia (*N* = 296)	
	*n*	%	*n*	%
Parent Age *				
<20	0	0	7	2.4
20–29	9	4.5	7	2.4
30–39	51	25.5	89	30.1
40–49	106	53.0	162	54.7
50–59	32	16.0	29	9.8
>60	2	1.0	2	0.7
Parent Education *				
None or primary	1	0.5	6	2.0
Secondary	59	29.5	13	4.4
TAFE ^†^	101	50.5	83	28.0
Undergrad	34	17.0	97	32.8
Postgrad	5	2.5	83	28.0
Child Gender *				
Male	163	81.5	195	65.9
Female	37	18.5	97	32.8
Child Age, mean (SD)	10.7	4.0	10.1	3.7
Location *				
City	79	39.5	141	50.0
Town	73	36.5	55	19.5
Regional ^‡^	6	3.0	58	20.6
Rural/Remote	42	21.0	28	9.9

^†^ TAFE: Technical and Further Education. ^‡^ Geographic regions outside major metropolitan areas. * Chi-squared test *p*-value < 0.05.

**Table 2 ijerph-18-11066-t002:** Diagnoses and perceived effect of the pandemic on symptoms changes in Italy and Australia.

	Italy (*N* = 200)			Australia (*N* = 296)		
	*n*	%	% with Worsening Symptoms	*n*	%	% with Worsening Symptoms
ADHD *	6	3.0	50.0	121	40.7	58.7
Anxiety Disorder	4	2.0	50.0	100	33.7	68.0
Autism Spectrum Disorder	102	51.0	37.3	181	61.3	58.0
Genetic Disorder	4	2.0	0	10	3.4	10.0
Intellectual or Learning Disorder	75	37.5	42.7	48	16.2	37.5
Obsessive–Compulsive Disorder	35	17.5	51.4	32	10.8	51.6
Tourette Syndrome	109	54.5	52.3	70	23.6	64.3

* Chi-squared test *p*-value < 0.05.

**Table 3 ijerph-18-11066-t003:** Caregivers’ mental health and wellbeing.

	Italy (*N* = 200)		Australia (*N* = 296)	
	*n*	%	*n*	%
Kessler *				
During the past 30 days, how often did you feel nervous?	72	36.0	60	20.3
During the past 30 days, how often did you feel hopeless?	50	25.0	33	11.2
During the past 30 days, how often did you feel restless or fidgety?	61	30.5	56	18.9
During the past 30 days, how often did you feel so depressed that nothing could cheer you up?	52	26.0	27	9.1
During the past 30 days, how often did you feel that everything was an effort?	40	20.0	76	25.7
During the past 30 days, how often did you feel worthless?	30	15.0	31	10.5

* Chi-squared test *p*-value < 0.05.

**Table 4 ijerph-18-11066-t004:** Linear regression results of NDDs on parental distress ^†^.

	Italy				Australia			
NDD	*B*	SE	R^2^	*F*	*B*	SE	R^2^	*F*
ASD	−2.363	1.958	0.099	4.131 *	0.329	0.679	0.112	5.916 *
ILD	−0.292	0.897			−1.985 *	0.831		
OCD	−0.134	1.228			0.998	1.072		
TS	2.709	2.083			2.416 *	0.872		

* *p* < 0.05, ^†^ After adjusting for the other NDDs, child age, caregivers’ age and educational level.

**Table 5 ijerph-18-11066-t005:** Linear regression results of the total sample *^,†^.

	Combined				Interaction Effect			
NDD	*B*	SE	R^2^	*F*	*B*	SE	R^2^	*F*
ASD	−0.428	0.643	0.168	14.640 *	−1.597	1.456	0.357	30.15 *
ILD	−1.266 *	0.682			1.961	1.864	0.326	26.54 *
OCD	0.294	0.485			−1.097	1.275	0.327	26.52 *
TS	2.761 *	0.845			0.314	0.605	0.362	30.77

* *p* < 0.05, ^†^ After adjusting for the other NDDs, child age, caregivers’ age and educational level.

**Table 6 ijerph-18-11066-t006:** Linear regression results of the symptomatic changes of specific NDDs on parental distress *^,†^.

	Italy				Australia			
NDD	*B*	SE	R^2^	*F*	*B*	SE	R^2^	*F*
ASD	−1.377	0.747	0.027	2.386 *	−1.167 *	0.395	0.059	4.47
ILD	−1.474	0.750	0.065	3.570	−2.085 *	0.943	0.130	3.295 *
OCD	−3.184	1.540	0.047	1.842	−0.095	0.889	0.163	2.885
TS	−1.628 *	0.615	0.052	3.983	0.329	0.5600	0.073	2.626

* *p* < 0.05, ^†^ After adjusting for caregivers’ age and educational level.

**Table 7 ijerph-18-11066-t007:** Linear regression results of the symptomatic changes of specific NDDs on parental distress *^,†^.

Combined				Interaction Effect			
B	SE	R^2^	F	B	SE	R^2^	F
−1.226 *	0.399	0.167	18.970 *	0.045	0.176	0.214	34.200 *
−2.349 *	0.587	0.166	9.007 *	−0.511	0.618	0.217	8.643 *
−1.607 *	1.392	0.475	20.320 *	3.002	2.853	0.223	14.160 *
−1.197 *	0.547	0.517	61.950 *	1.343	0.953	0.546	52.360 *

**p* < 0.05, ^†^ After adjusting for caregivers’ age and educational level.

## Data Availability

The dataset has been uploaded to a public database: doi: 10.6084/m9.figshare.16528998.

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
