# Peer review of "Distress Levels of Parents of Children with Neurodevelopmental Disorders during the COVID-19 Pandemic: A Comparison between Italy and Australia"

_ijerph, 2021, doi:10.3390/ijerph182111066_

Round 1

Reviewer 1 Report

In a period in which COVID 19 pandemic marked the lives of people globally, this paper analyzes the impact on a specific population groups assuming the challenge to compare  two countries at the antipodes.

The authors investigate the psychological impact of COVID 19 pandemic on the parents and caregivers of children with neurodevelopmental disorders, with focus on various aspects such as general wellbeing, family health, home-based learning and child behaviors and emotions. The comparison made between Italy and Australia is interesting considering that they were so different in terms of the evolution of the pandemic and I appreciate that the authors considered in this regard an objective indicator (the stringency index values).

The article is well organized and contains all the sections, which are well developed. However, some minor revisions are needed.

Introduction- is clearly and coherently presented and supported by the latest literature in the field

Methods – the authors should describe the selection procedure of the participants  

   - Please briefly describe the standardized instrument (The 6-item Kessler 103 Psychological Distress Scale – K6) and mention its author (including in the references list)

Results – at first sight a little more difficult to follow, especially since it seems to me that there are some errors in numbering some tables and figures

Page 4, Table 1 – authors should specify in a legend what means “TAFE”   

                          -please specify    what the term "regional" refers to

Page 5-6, lines 166-167 - The sentence “Surprisingly, respondents in Australia stated that nearly more and less medication is normal, 97.07% and 98.74%, respectively” is, in my opinion, confusing and can be understood after consulting the supplementary material. The authors should be more explicit and rephrase it.

Page 6, line 196 - the authors should verify the mention of table 5; it seems to be table 4

Page 6, lines 207-210 and figure 1 (page 7) – according to the figure 1 it seems that in Australia the main contributor to parental distress is TS;  the authors should clarify this

Page 7, line 227 – authors should verify if it is “figure 2”; it seems to be “figure 3”

Page 8, lines 236-237 and lines 240-243 - they seem to be the legend from the figure 2 and 3, respectively; maybe it would be better to write this text with a smaller font.

Discussions - Page 9, lines 266-271 - the sentence is too long and a little unclear. Furthermore, the results (the findings) presented do not refer to the parent-child relationship and do not support the statement that “predictors for adverse mental distress in Australian parents included .. poorer quality of parent-child relationship”

Page 9 -The following sentences seem to be contradictory :  “Notably, the diagnosis of TS in children appeared to be associated with lower levels of parental distress in families with children diagnosed with other NDDs” (lines 249-251)   AND  “In addition, our study found that among children with NDDs, those with TS had the  most significant impact on parental distress” (lines 272-273). The authors should clarify this aspect.

Page 9 -The following sentence  “..our study did not show any significant results with regards to children diagnosed with ASD increasing parental distress during the pandemic” (lines 281-283) it seems to be different and does not reflect what is presented in the Results section “Although there were no significant associations between NDDs and parental distress in Italy, children with ASD showed a positive correlation with parental distress” (lines 201-203). The authors should clarify this aspect, too. 

There are many results obtained from the survey and presented on pages 5-6 about which the authors do not mention anything in the Discussion section. Authors should include some comments (a brief paragraph) on these results (for example about the aspect that “On average, parents in Italy showed higher distress levels compared to parents in Australia” and  meanwhile parents in Australia perceived/ self-reported  a higher impact on most of the aspects considered in this research).  These results should be briefly commented on, especially as the title of the article refers to comparisons between countries. Perhaps these comments can be linked with some of the study limitations related to cross-country comparisons, which authors have already presented in lines 296-307. The results of this study can also be a starting point for future research to identify other factors that may influence the perceived impact of COVID 19 pandemic on parents with children with neurodevelopmental disorders.

References list – please, verify the references no 19 (a few extra words appear), no 9 and 11 (2021 Dec means 2021 December?)

Date: 5 Sep 2021

Reviewer 2 Report

Abstract:
Clearly point out the purpose of the study
Redo the method part of the summary, indicating at least: Indicate type of study, Indicate instrument names, N, etc.
Redo results (indicate main findings) and conclusion (respond to objective)

Introduction:
Improving this section: it needs more grounding to discuss the importance of the theme and the gap in the literature. Indicate in the last paragraph the objective.

Methods
Have the instruments been validated? If so, indicate their validation reference.
If not, there is a measurement bias and difficult to correct.
Who collected? Who answered for the children?
Was there approval by the human ethics committee? Did children and adolescents agree to participate? What gender are the volunteers? What regions were they from? How was the sample (convenience or was there a sample calculation?)
How was the regression analysis done? How were the models made? What significance level used to insert the variables into the model? Describe in more detail the statistical analysis.

Results:
Show in tables 1-4 the comparison test between countries. (chi-square)
Figure 3 used which comparison test?

Discussion:
The impact of the pandemic in the 2 countries was very different, this should be pointed out in the discussion with greater wealth of data, both with regard to victims, as well as cases, lockdwns and changes in routines, in addition, an analysis of the situation of children (culturally speaking) in a pre-pandemic state, as they are countries from different continents and very different cultures as well.

Conclusion:
OK

Round 2

Reviewer 3 Report

Thank you for the opportunity to review a revision of the paper ‘Distress levels of parents of children with 2 neurodevelopmental disorders during the COVID-19 3 pandemic: a comparison between Italy and Australia’. I acknowledge and appreciate the authors’ responsiveness to the earlier reviews. This paper is more coherent and internally consistent, and makes its point in a much more cogent manner. The abstract is markedly improved. There remain a few issues to consider and resolve.

Line 36: ‘Although this’—the antecedent is unclear; please specify ‘this what’

Line 67: ‘the lock down’— I would suggest a more general ‘lockdowns’ (without ‘the’) since these varied throughout Australia

Line 145: ‘using the mixed model’— since the version of mixed model is unclear, I would say ‘a mixed model analysis’ or similar

Table 1 Commented note: the chi-square statistic is unclear, reports only the whole number, and it is not clear to what this comment attaches. Please clarify and delete the comment.

Line 178: ‘While in Australia…’—omit ‘While’ since this is not a sentence as it stands.

Line 187 ‘higher worsening of symptoms’—this is an odd way to phrase this idea, and it is unclear. Consider ‘increased symptoms’ or similar.

Line 201: ‘in Australia, 25.68%’—has two decimal places

Line 219: ’97.1% of respondents’—begins a sentence with a numeral. Reword the sentence.

Section 3.2: This is a linear regression analysis that should typically report correlations r, r2, F values and/or β statistics. Throughout this section and in Tables 4 and 5 only an unspecified ‘estimate’ is reported. This section and these Tables should conform to standard statistical reporting.

Figures 1,2,3 are in colour. This is fine for online access, but may not be so legible in black and white. I leave this issue to the journal editors.

Line 376: in the limitations section the authors note that a self-report bias cannot be excluded. This is fine (although I would say ‘parent reports cannot be verified’, but that is not essential), however a key limitation here is that this was a self-selecting sample of convenience, and that is also a key issue. As I noted in my first review, any number of elaborate statistical analyses cannot make up for data which may be problematic from the very beginning. There may have been some reason why parents of some children with particular conditions chose to participate and some did not, and elegant analysis cannot compensate for that, and conclusions may be fatally impaired. The self-selection limitation must be acknowledged.

These are relative minor issues to resolve and I hope the authors will persist with the editing. It is a much improved paper that posits some tentative conclusions. Rather than simply saying ‘further research is warranted’ in the Conclusion, I wonder if the authors might want to suggest how that research might be carried out, particularly in terms of sample recruitment, in order to propose how their findings might be confirmed? That is a question, not a requirement.
